# Disparities in Medicaid and Medicare physician reimbursements for ophthalmic procedures

Angela R. Elam [1,2,3]*, Dennis Akrobetu [4], Tochukwu Ndukwe [5], David Sidhom [6], Patrice M. Hicks [1], Shikha Marwah [1], Brian C. Stagg [7], Paul P. Lee [1,2,3], Josh D. Stein [1,2,3], Paula Anne Newman-Casey [1,2,3]

1 Department of Ophthalmology and Visual Sciences, W.K. Kellogg Eye Center, University of Michigan, Ann Arbor, Michigan, 2 Institute for Healthcare Policy and Innovation, University of Michigan, Ann Arbor, Michigan, 3 Center for Eye Policy and Innovation, University of Michigan, Ann Arbor, Michigan, 4 Department of Ophthalmology, Massachusetts Eye and Ear, Harvard Medical School, Boston, Massachusetts, 5 Department of Ophthalmology, University of Illinois College of Medicine, Chicago, Illinois, 6 University of Michigan Medical School, Ann Arbor, Michigan, 7 Department of Ophthalmology, John A. Moran Eye Center, University of Utah, Salt Lake City, Utah

* aelam@med.umich.edu

## Introduction

Medicare and Medicaid are government-sponsored programs that provide insurance to 14.2% and 19.8% of the United States population, respectively [1,2]. Currently, of the 328 million people in the US, there are 59 million people insured by Medicare and 72 million insured by Medicaid [3]. As two of the largest insurers in the country, the policies that both programs establish have significant implications for overall access to healthcare in the United States. In addition, these two programs, especially Medicare, help set precedent and care reimbursement levels for other insurers. Medicare is a federally-run program whose primary role is to provide insurance to US adults over the age of 65 and some people with disabilities [1,4]. In contrast, Medicaid operates at both the federal and state levels and provides medical coverage primarily to low-income individuals, children, people with disabilities, and low income [5].

When comparing physician reimbursement under both programs, Medicaid has historically reimbursed physicians at about 60–80% of Medicare rates [6]. However, the difference between Medicaid and Medicare reimbursement varies dramatically for highly specialized and technical services such as radiographic studies and surgical procedures [7–10]. In addition, each state is responsible for determining Medicaid payment levels, which can lead to inter-state variability in Medicaid reimbursement [7]. This variability may lead to geographic disparities in healthcare access across states for people insured by Medicaid.

Demonstrating a connection between Medicaid reimbursement and ophthalmologists' willingness to provide care to patients with Medicaid insurance may be helpful for health policy advocacy efforts by state and national professional ophthalmic societies, such as the American Academy of Ophthalmology (AAO). Physicians are more reluctant to accept Medicaid patients when reimbursement is not deemed adequate [7,11,12]. Medicaid patients are less successful at obtaining eye care appointments than privately-insured patients [13]. Lower physician reimbursement for Medicaid

**Data availability statement:** All relevant data are within the manuscript and its Supporting Information files.

**Funding:** Financial Support: This work was supported by National Institute for Minority Health and Health Disparities (Bethesda, MD; K23MD016430, ARE), National Institute on Aging/Michigan Center for Urban African American Aging Research (Bethesda, MD; P30AG015281, ARE), National Eye Institute (Bethesda, MD; R01EY031337, PANC), and Research to Prevent Blindness (Career Development Award, PANC; Unrestricted Grant ARE, PPL, BS). The funding organization had no role in the design and conduct of the study; collection, management, analysis, and interpretation of the data; preparation, review, or approval of the manuscript; and decision to submit the manuscript for publication.

**Competing interests:** The authors have declared that no competing interests exist.

patients could exacerbate disparities in access and quality of eye care delivered to particularly vulnerable patients. Other studies have shown that increased Medicaid reimbursement is associated with greater physician willingness to accept patients with Medicaid insurance [7,14,15].

No prior studies have investigated the differences between Medicare and Medicaid physician reimbursement nor inter-state variability in Medicaid reimbursement for ophthalmic procedures and how these differences may correlate to ophthalmologists' acceptance of patients with Medicaid insurance. Thus, the objectives of this study were to 1) directly compare Medicare and Medicaid physician reimbursement rates across the United States, 2) quantify existing inter-state variation between Medicare and Medicaid reimbursement for commonly performed ophthalmic procedures, and 3) identify any correlation between state Medicaid reimbursements rates and ophthalmologists' acceptance of patients with Medicaid insurance within that state.

## Methods

This is a retrospective cross-sectional study using the most recent data available (2021) on Medicaid and Medicare reimbursement fees for common ophthalmic procedures in the 49 US states and territories (48 states plus Washington D.C.) that met inclusion criteria. This study was reviewed by the Institutional Review Board at the University of Michigan and found to not fit the

definition of human subjects research requiring IRB approval. All states that participated in Medicare and Medicaid with fee-for-service reimbursement were included in the analysis. The only two states excluded were Kansas and Tennessee since these two states do not have a fee-for-service system of reimbursement for Medicaid, using managed care organizations [16].

### Ophthalmic procedures

Eighteen commonly performed ophthalmic procedures were identified by the study team. These procedures were chosen to include one or more common procedures from each of the various surgical subspecialties in ophthalmology performed both in the operating room (OR) and in the office. The procedures included were: 1) comprehensive ophthalmology, cataract (OR), open globe repair (OR), punctal plug placement (office), YAG capsulotomy (office); 2) glaucoma, glaucoma tube implant (OR), trabeculectomy (OR), gonioscopy (office), laser trabeculoplasty (office); 3) retina, vitrectomy (OR), intravitreal injection (office), B scan ultrasonography (office); 4) cornea, penetrating keratoplasty (OR), corneal foreign body removal (office); 5) pediatrics and strabismus, strabismus surgery (OR); 6) oculoplastics, ptosis repair (OR), enucleation (OR), nasolacrimal probing (office). Each procedure's Current Procedural Terminology (CPT) code was identified using the CPT 2021 Professional Edition. For procedures with more than one CPT code, the code with the broadest scope was chosen. The procedures were analyzed individually and as two larger groups, classified by the site at which they are predominantly performed (in-office and OR procedures).

## Fee schedules for reimbursement

The global fee schedule reimbursements, calculated in US dollars, for Medicaid and Medicare were analyzed. The global fee is the sum of the fees of two components—the Technical Component (TC) and the Professional Component (PC). The Medicaid and Medicare fee schedules were identified using publicly-available data (S1_Supplemental Table 1). These data sources are the minimal data sets used for this study. The Medicaid fee schedule was obtained from each individual state's Medicaid website. If the fee schedule was not available on the state's Medicaid website, it was obtained by emailing or calling the respective state's Medicaid department between March and May 2021. Medicare fee schedules were identified using the Centers for Medicare and Medicaid Services Physician Fee Schedule search tool [17] (accessed on March 24, 2021). In states where more than one Medicare Administrative Contractor was listed for reimbursement, the region listed as "rest of state" was selected in order to use the lower Medicare reimbursement rate, as this selection would bias the hypothesis that Medicare has higher reimbursement towards the null [7].

## American academy of Ophthalmology membership database

In September and October 2021, the 2021 AAO Practice Environment Member Survey was administered to a randomized sample of U.S. practicing ophthalmologists who were members of the American Academy of Ophthalmology (AAO). There was a 10% overall response rate. The respondents were overall representative of the AAO membership, in terms of practice type, primary focus, age, practice group type, and gender. The single item regarding Medicare and Medicaid reimbursement question posed in the survey was "What is your current position on accepting Medicaid and Medicare patients?" Respondents were asked to respond based on their practice from 3 options: "Accept all patients", "Limit the number of patients", "Do not accept any patients." These 3 options were given for Medicare and Medicaid. These respondents were grouped by location into five regions of the United States specified by the AAO: West (Montana, Wyoming, Colorado, Utah, Nevada, Idaho, Washington, Oregon, California, Alaska, Hawaii), Midwest (Iowa, Michigan, Nebraska, Minnesota, Wisconsin, Illinois, Indiana, Ohio, North Dakota, South Dakota, Missouri), Southwest (Texas, Oklahoma, New Mexico, Arizona), Southeast (Arkansas, Florida, Louisiana, Mississippi, Kentucky, North Carolina, South Carolina, Maryland, Delaware, West Virginia, Virginia, D.C., Georgia, Alabama), and Northeast (Pennsylvania, New York, Vermont, Maine, New Hampshire, Rhode Island, Connecticut, New Jersey, Massachusetts).

## Statistical analysis

The differences in the dollar amount of reimbursement for Medicare and Medicaid were compared by procedure for each state and between states. For each procedure, we calculated the mean, standard deviation, median, and interquartile range of dollar amount differences across all included states for both Medicaid and Medicare reimbursements. The coefficient of variation (CV), a measure that is used to quantify variability, was calculated by dividing the standard deviation by the mean reimbursement for each procedure. Finally, for each procedure we tested whether the average Medicaid and Medicare reimbursements are statistically equal, using a two-sided paired t-test. A p-value $< 0.05$ was considered statistically significant. All analyses were carried out with the statistical analysis software R, version 4.1.0 (R Foundation for Statistical Computing, Vienna, Austria).

## Results

The average Medicaid reimbursement rate for both OR and in-office procedures varied much more than the average Medicare reimbursement rate across states (Fig 1). The average Medicaid reimbursement rate is lower than the average Medicare reimbursement rate for OR procedures ($p < 0.001$) and in-office procedures ($p < 0.001$). The average Medicaid reimbursement for OR procedures was lower than the average Medicare reimbursements in 42 of the 49 (85.7%) states and territories studied. In the case of in-office procedures, the average Medicaid reimbursement was lower in 36 out of the

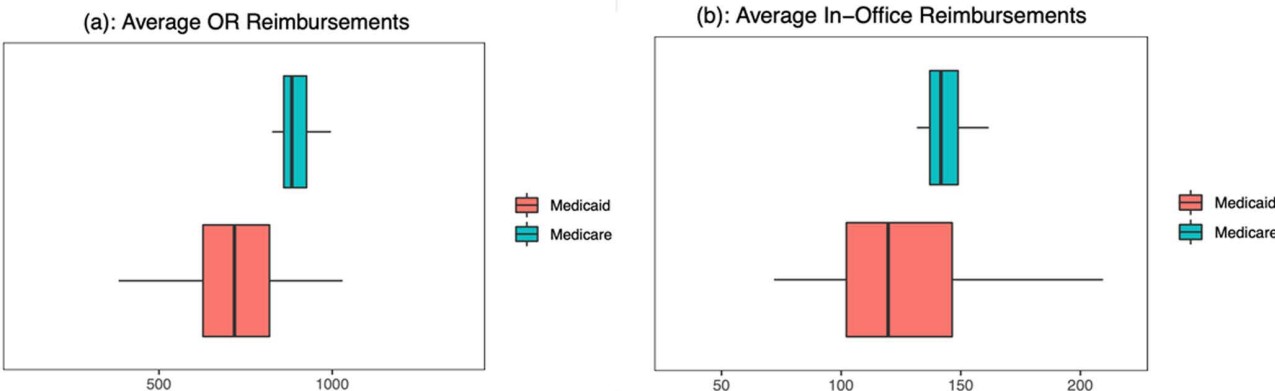

**Fig 1. Average OR and in-office procedure reimbursements.**

49 (73.5%) states and territories included in the study. There are some states (Alaska, Arizona, Montana, South Dakota, Nebraska, and Arkansas) that on average reimburse physicians more for Medicaid than Medicare for OR (Fig 2) and in-office ophthalmic (Fig 3) procedures.

Comparing the dollar difference in Medicaid and Medicare reimbursement by procedure, we find that the largest positive dollar difference for OR procedure reimbursements was for enucleation with Medicaid paying $2106.94 (USD) more in Rhode Island than Medicare, and the largest negative difference was for open globe repair in New Jersey with

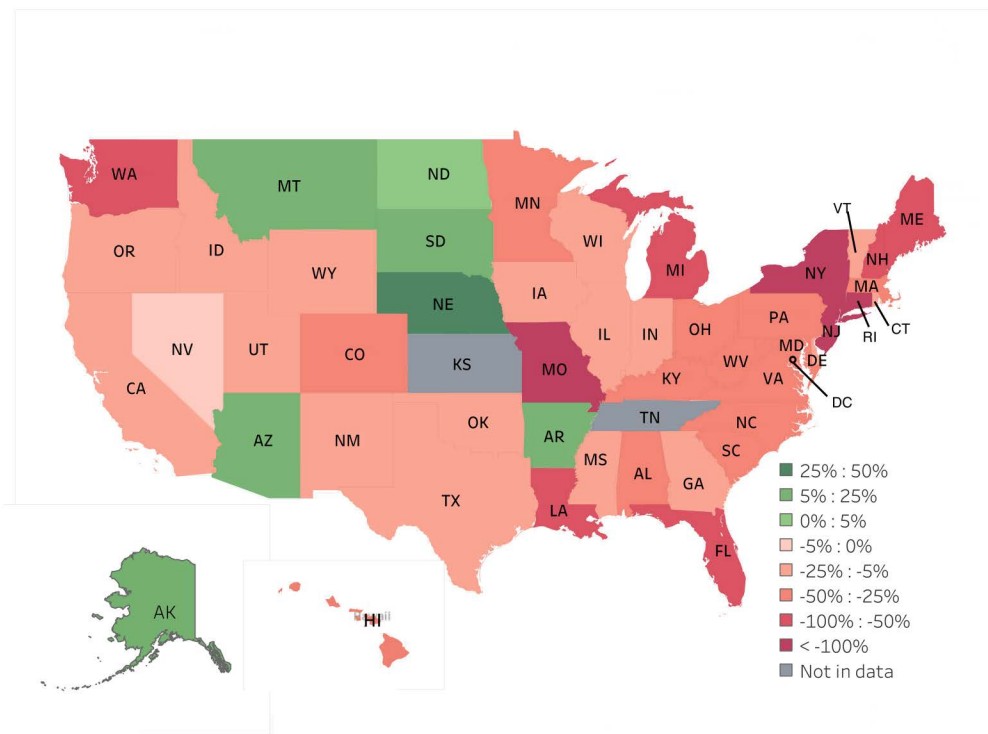

**Fig 2. % Differences Between Medicaid and Medicare for OR procedures.**

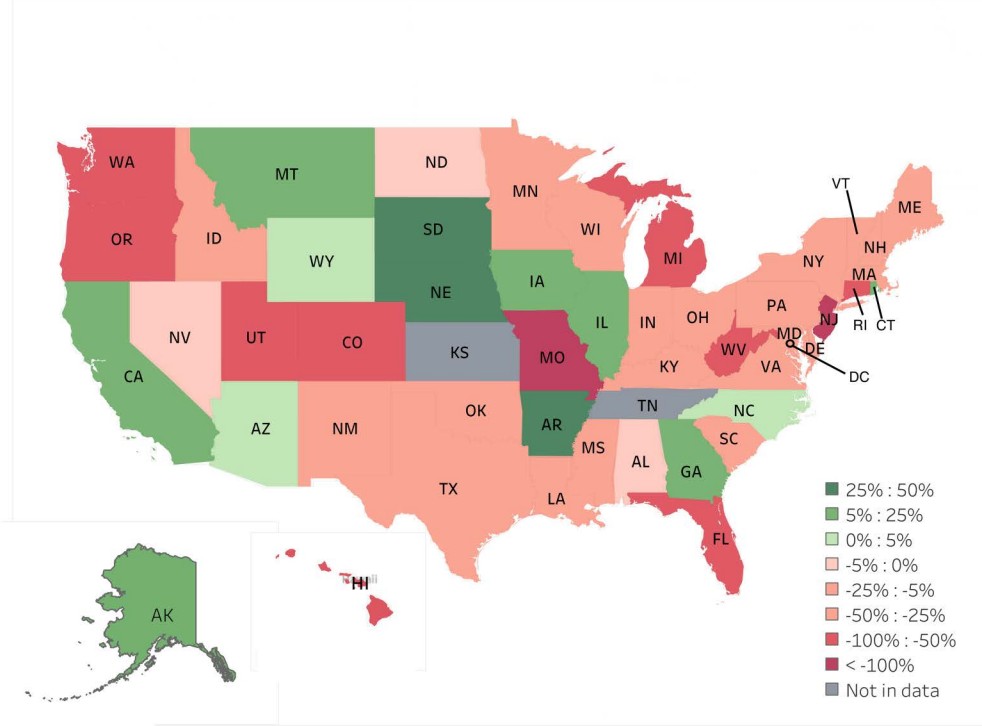

**Fig 3.  % Differences Between Medicaid and Medicare for in-office procedures.**

Medicaid paying $998.78 less then Medicare. For in-office procedures, the largest positive dollar difference for reimbursements was for laser trabeculoplasty with Medicaid paying $375 more in South Dakota than Medicare. The largest negative dollar difference was for YAG capsulotomy with Medicaid paying $272.25 less than Medicare in Missouri. The mean dollar difference in Medicaid and Medicare reimbursement is negative for all OR procedures except cataract surgery, meaning Medicaid reimburses less than Medicare for the majority of these procedures. The same is true for all except three in-office procedures (intravitreal injection, laser trabeculoplasty, and ultrasound b-scan) (Table 1).

Mean Medicaid reimbursement for all investigated OR procedures varied from $113.53 in Missouri to $1377.24 in Alaska. Mean Medicaid reimbursement for in-office procedures varied from $31.33 in Missouri to $218.78 in Nebraska. Mean Medicare reimbursement for OR procedures varied from $826.46 in Arkansas to $1164.75 in Alaska. In the case of in-office procedures, the Medicare mean reimbursement varied from $131.67 in Arkansas to $183.66 in Alaska.

Enucleation (CV = 0.58) was the OR procedure with the highest Medicaid reimbursement variation between states, whereas glaucoma tube implant (CV = 0.29) was the OR procedure with the lowest Medicaid reimbursement variation between states. Intravitreal injection (CV = 0.62), was the in-office procedure with the highest Medicaid reimbursement variation between states, whereas punctal plug (CV = 0.36) was the in-office procedure with the lowest Medicaid reimbursement variation between states (Table 2).

The AAO survey data show that across the five regions of the country, the percentage of ophthalmologists respondents that accept all patients with Medicaid in their practice ranges from 56%, in the Northeast and Southeast regions, to 66% in the Midwest (Fig 4, Table 3). In contrast, we find that 95% of ophthalmologists who responded to the survey report accepting patients who have Medicare (Fig 5). The median dollar reimbursement difference between Medicaid and Medicare also varies significantly across regions, ranging from a difference of $86.85 in the Southwest to $293.72 in the Northeast for OR procedures and $9.59 (Southwest) to $35.71 (Northeast) for in-office procedures (Table 4). When stratified by US

**Table 1. Dollar difference between medicaid and medicare mean reimbursements (across states). for OR and In-office Procedures.**

| Procedures | Difference | |
|---|---|---|
| | Mean (SD) | Median (Range) |
| Cataract surgery | 18.82 (236.48) | −4.71 (−94.71: 59.89) |
| Glaucoma tube implant | −266.69 (232.98) | −265.1 (−367.48: −105.95) |
| Trabeculectomy | −272.69 (235.31) | −270.14 (−398.04: −123.92) |
| Vitrectomy | −87.93 (340.07) | −139.55 (−233.15: −50.17) |
| Penetrating Keratoplasty | −264.14 (306.17) | −274.08 (−426.19: −116.8) |
| Open globe repair | −307.43 (254.81) | −286.06 (−433.35: −185.43) |
| Strabismus surgery | −118.48 (151.38) | −125.74 (−206.2: −55.05) |
| Ptosis Repair | −21.1 (218.65) | −18.53 (−154.58: 61.51) |
| Enucleation | −222.92 (406.07) | −272.35 (−397.68: −142.37) |
| Intravitreal injection | 20.26 (82.86) | −9.92 (−29.63: 44.45) |
| Punctal plug | −45.39 (37.22) | −38.16 (−68.74: −24.42) |
| Chalazion removal | −36.76 (34.05) | −34.32 (−55.76: −16.69) |
| Gonioscopy | −6.14 (7.69) | −7.36 (−11.31: −2.57) |
| YAG capsulotomy | −78.8 (92.66) | −88.91 (−136.75: −29.39) |
| Corneal foreign body removal | −11.24 (25.59) | −18.25 (−27.04: −4.54) |
| Nasolacrimal probing | −30.45 (49.60) | −31.3 (−61.86: 5.03) |
| Laser trabeculoplasty | 7.79 (125.52) | −17.73 (−66.47: 31.91) |
| B-scan ultrasonography | 13.91 (41.57) | 7.42 (−16.36: 34.73) |

regions, Medicaid acceptance rates varied by region and appeared to have correlation with reimbursement rates. For example, the Southeast and Northeast regions have low acceptance rates for Medicaid, and we observe that the dollar difference in Medicaid and Medicare reimbursement is greatest in those regions.

## Discussion

Significant differences in Medicaid and Medicare reimbursements exist for ophthalmic procedures. Additionally, Medicaid reimbursement overall is significantly lower than Medicare for surgical and outpatient ophthalmic procedures. The difference between Medicaid and Medicare reimbursements is not statistically significant for cataract surgery, vitrectomy, ptosis repair, and laser trabeculoplasty. The range of reimbursements by state varied widely with these procedures with very high reimbursement outliers for some states, such as Nebraska, which decreases the confidence interval and affects statistical significance. This study supports the widespread impression that ophthalmologists are compensated at lower reimbursements when taking care of people insured by Medicaid compared to when taking care of people insured by Medicare. The lower ophthalmologist reimbursement rates may disincentivize physicians from providing eye care for patients with Medicaid insurance.

There is also substantial inter-state variation for Medicaid reimbursements. For example, the CV for Medicaid varied widely and was always at least 0.29, with some procedures, such as intravitreal injections, having a CV as high as 0.62. This high CV reflects the large variability in Medicaid reimbursement for intravitreal injections between states, with fees ranging from $56.39 in Florida to $443.67 in Nebraska. In comparison, the CV for Medicare was 0.07 for all states, which is lower than Medicaid. Prior studies in orthopedic surgery [16] and radiology [7] found similar rates of variation in Medicaid reimbursements with CV ranges of 0.32–0.57 and 0.26–0.62, respectively. An investigation of disparities in Medicaid physician reimbursements for vascular surgery also found significant variation [9]. Again, ophthalmologists, alongside other physicians from a variety of different specialties, are receiving disparate reimbursements for Medicaid patients depending on the state in which they practice.

**Table 2. Descriptive Statistics and Paired t-test p value for Medicare and Medicaid Reimbursements.**

| Procedure | Medicare Reimbursements | | | Medicaid Reimbursements | | | Test |
| --- | --- | --- | --- | --- | --- | --- | --- |
| | Mean (SD) | CV | Median (Range) | Mean (SD) | CV | Median (Range) | p value |
| Cataract surgery | 539.3 (36.04) | 0.07 | 529.43 (516.03 - 554.11) | 558.13 (235.45) | 0.42 | 513 (435.09 - 602.53) | 0.58 |
| Glaucoma tube implant | 1127.59 (75.93) | 0.07 | 1104.92 (1078.88 - 1160.24) | 860.9 (248.55) | 0.29 | 856.2 (737 - 981.14) | <0.001 |
| Trabeculectomy | 1082.91 (73.23) | 0.07 | 1062.55 (1034.17 - 1114.48) | 810.22 (244.73) | 0.30 | 814.69 (664.45 - 928.62) | <0.001 |
| Vitrectomy | 888.26 (59.44) | 0.07 | 869.87 (850.13 - 913.01) | 800.34 (340.80) | 0.43 | 747.38 (658.37 - 828.68) | 0.08 |
| Penetrating Keratoplasty | 1241.95 (83.57) | 0.07 | 1219 (1187.26 - 1277.47) | 977.81 (313.81) | 0.32 | 961.26 (815.06 - 1116.9) | <0.001 |
| Open globe repair | 1097.75 (72.96) | 0.07 | 1078.13 (1050.97 - 1127.15) | 790.33 (266.14) | 0.34 | 826.3 (644.63 - 915.54) | <0.001 |
| Strabismus surgery | 591.96 (39.86) | 0.07 | 581.01 (565.85 - 608.94) | 473.48 (154.41) | 0.33 | 467.89 (388.49 - 521.22) | <0.001 |
| Ptosis Repair | 589.31 (39.43) | 0.07 | 578.72 (563.65 - 605.8) | 568.21 (223.28) | 0.39 | 553.3 (466.83 - 625.92) | 0.50 |
| Enucleation | 939.19 (66.28) | 0.07 | 921.39 (890.82 - 969.31) | 716.27 (416.20) | 0.58 | 652.45 (527.63 - 778.79) | <0.001 |
| Intravitreal injection | 112.65 (8.05) | 0.07 | 110.8 (107.58 - 115.06) | 132.91 (82.13) | 0.62 | 103.53 (83.34 - 161.62) | 0.09 |
| Punctal plug | 149.29 (10.81) | 0.07 | 146.67 (141.8 - 153.83) | 103.89 (37.61) | 0.36 | 109.49 (80 - 125.88) | <0.001 |
| Chalazion removal | 128.25 (8.97) | 0.07 | 125.75 (121.83 - 132.44) | 91.49 (34.35) | 0.38 | 93.09 (68.88 - 107.28) | <0.001 |
| Gonioscopy | 27.89 (1.89) | 0.07 | 27.44 (26.62 - 28.45) | 21.75 (8.05) | 0.37 | 20.99 (16 - 24.49) | <0.001 |
| YAG capsulotomy | 332.04 (23.46) | 0.07 | 325.63 (315.12 - 340.24) | 253.24 (94.64) | 0.37 | 244.12 (191.43 - 292.54) | <0.001 |
| Corneal foreign body removal | 67.92 (4.64) | 0.07 | 66.65 (64.69 - 69.85) | 56.68 (25.76) | 0.45 | 52.46 (41.68 - 64.13) | <0.001 |
| Nasolacrimal probing | 161.76 (11.66) | 0.07 | 158.69 (153.04 - 167.25) | 131.31 (50.20) | 0.38 | 130.3 (103.5 - 172.57) | <0.001 |
| Laser trabeculoplasty | 246.01 (16.83) | 0.07 | 241.24 (234.45 - 253.54) | 253.8 (124.68) | 0.49 | 232.84 (188.8 - 276.74) | 0.67 |
| B-scan ultrasonography | 72.92 (5.29) | 0.07 | 71.63 (69.26 - 74.83) | 86.84 (41.70) | 0.48 | 79.95 (60.39 - 109.69) | 0.02 |

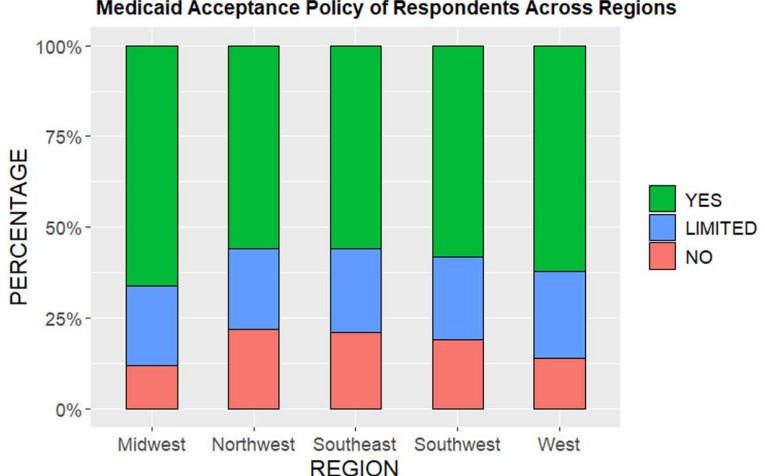

Number of Respondents by Region: Midwest 197, Northeast 179, Southeast 236, Southwest 99,

**Fig 4. Medicaid Acceptance Policy of AAO Respondents by Region.**

The Medicaid and CHIP Payment and Access Commission [18] found that physicians were significantly less likely to accept new patients insured by Medicaid (74% acceptance) compared to those insured by Medicare (88%). Physicians caring for patients at family planning clinics (100%) and community health centers (97.3%) are most likely to accept new Medicaid patients, compared to those in private practice (69.9%).[18] As such, it is likely that one of the biggest factors influencing physician willingness to accept Medicaid patients is physician reimbursement and the practice context.

**Table 3. Medicaid and Medicare acceptance policy, by region.**

| Region | Medicaid | | Medicare | | |
|---|---|---|---|---|---|
| | Accept | Do not/Limited | Accept | Do not/Limited | Total |
| West | 107 (62%) | 66 | 159 (92%) | 14 | 173 |
| Midwest | 131 (66%) | 66 | 193 (98%) | 4 | 197 |
| Southwest | 57 (57%) | 42 | 88 (89%) | 11 | 99 |
| Southeast | 133 (56%) | 103 | 223 (94%) | 13 | 236 |
| Northeast | 100 (56%) | 79 | 174 (97%) | 5 | 179 |
| U.S. | 528 (60%) | 356 | 837 (95%) | 47 | 884 |

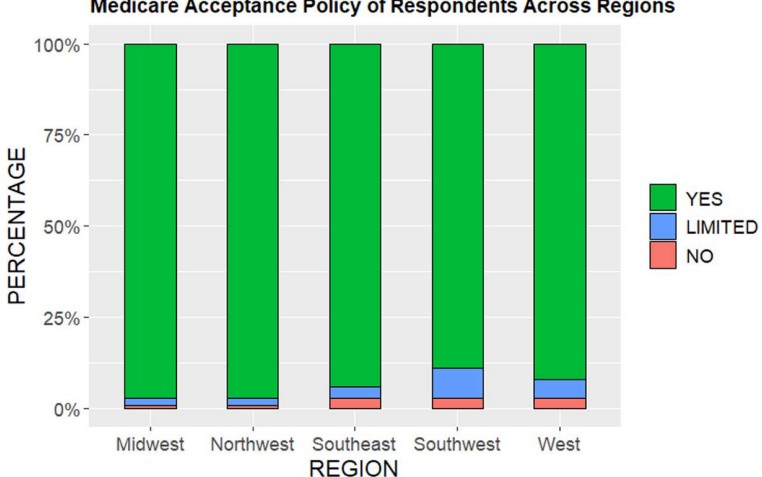

Number of Respondents by Region: Midwest 197, Northeast 179, Southeast 236, Southwest 99,

**Fig 5. Medicare Acceptance Policy of AAO Respondents by Region.**

**Table 4. Median difference between Medicaid and Medicare dollar reimbursements for the investigated OR and In-Patient procedures, by region.**

| Region | Median Reimbursement Difference (Medicaid minus Medicare, in dollars) | |
|---|---|---|
| | OR Procedures | In-Office Procedures |
| West | −131.27 | −12.58 |
| Midwest | −129.47 | −11.88 |
| Southwest | −86.85 | −9.59 |
| Southeast | −219.63 | −22.37 |
| Northeast | −293.72 | −35.71 |

According to the 2021 AAO Practice Environment Member Survey data, only 60% of ophthalmologists accept patients with Medicaid insurance, compared to 95% that accept Medicare. While our data do not allow us to demonstrate a statistically significant correlation, we do find that acceptance rates of Medicaid vary by region and appear to correlate with reimbursement rates. Regions with lower Medicaid acceptance rates also have greater dollar difference in Medicaid and Medicare reimbursement. In the Midwest region, the Medicaid acceptance rate is relatively higher, and we notice the reimbursement difference between Medicaid and Medicare is lowest in that region, meaning the Medicaid reimbursement is higher (closer

to that of Medicare). Because Medicaid policy is often determined at the state level, it would likely benefit the AAO and state ophthalmic societies to identify any correlation between Medicaid acceptance and physician reimbursement rates, as this information is critical for informing Medicaid policies.

Our data show significant differences between states for Medicaid reimbursements, as well as significant differences between Medicare and Medicaid reimbursements for individual states. Because lower Medicaid reimbursements may contribute to disparities in eye care for Medicaid patients [19], future work will focus on assessing whether the inter-state variability in Medicaid reimbursement rate is associated with better vision outcomes or access to care in states with higher reimbursement rates and worse vision outcomes or access in states with lower reimbursement rates. While White Americans make up the largest racial/ethnic group with Medicaid insurance, racial and ethnic minorities are disproportionately represented amongst Medicaid enrollees and everyone with Medicaid has lower socioeconomic status (SES) than most Americans [20]. The very same populations (those who live in poverty and those who identify as racial and ethnic minorities) are at higher risk of underutilization of eye care and bear a disproportionate share of the burden of vision loss in the US [21,22]. Addressing how to remove systems-level barriers to accessing eye care is highly relevant to health policy efforts to address disparities in eye care and promote care for those most at risk for vision loss.

While there are national guidelines regarding who qualifies for Medicaid and what medical services they are offered, there are no national guidelines to define physician reimbursement for services from Medicaid. In contrast, the framework for Medicare physician reimbursement rates is set by Centers for Medicare and Medicaid Services nationally. Medicare compensation structure is intricate and accounts for geographic variation, but the payments reflect the differential costs of resources in different geographic locations and are so fine-tuned that they are often used as benchmarks by commercial insurance companies. Though federal provisions were made to require that Medicaid payments be sufficient to ensure equal access to care for Medicaid patients, there is no regulatory process – such as the one in place for Medicare – to ensure the successful implementation of this provision [23].

This study has several limitations. The data from all 50 states in the US were not available. However, the data available incorporates over 97% of the US population in our analyses [24]. While there are numerous outpatient and surgical ophthalmic procedures performed, we have only analyzed 18 procedures. The goal was to choose a variety of representative ophthalmic procedures, though we acknowledge that differences in reimbursement found in this study may be different when analyzing other procedures. With regards to demonstrating statistically significant correlation between Medicaid reimbursement and acceptance rates, doing so would require more data points, so doing this at the state or county level (as opposed to regions) would likely allow for those analyses. Unfortunately, we were unable to analyze the AAO survey data at a more granular level.

Physician reimbursement rates are significantly lower for Medicaid compared to Medicare reimbursement rates and vary substantially by state for Medicaid. Given that some of our country's most vulnerable populations' health and health care relies upon Medicaid, it is important that we learn more about how these disparities may affect access to and quality of eye care. This and additional granular work at individual state and national levels will be critical in informing health policy to ensure that systemic factors, such as physician reimbursement, are not contributing to disparities in eye care and hindering our goal of reaching equity in vision and eye care.

## Supporting information

**S1 Supplemental Table 1.  Medicare and Medicaid Data Sources.**
(DOCX)

## Author contributions

**Conceptualization:** Angela Elam, Brian C. Stagg.

**Data curation:** Angela Elam, Dennis Akrobetu, Tochukwu Ndukwe, David Sidhom, Shikha Marwah.

**Formal analysis:** Angela Elam, Patrice M. Hicks, Shikha Marwah, Paula Anne Newman-Casey.

**Funding acquisition:** Angela Elam.

**Investigation:** Angela Elam, Dennis Akrobetu, Tochukwu Ndukwe, David Sidhom.

**Methodology:** Angela Elam, Dennis Akrobetu, Tochukwu Ndukwe, David Sidhom, Patrice M. Hicks, Shikha Marwah, Brian C. Stagg.

**Project administration:** Angela Elam, Paula Anne Newman-Casey.

**Resources:** Angela Elam.

**Supervision:** Angela Elam, Patrice M. Hicks, Paul P. Lee, Paula Anne Newman-Casey.

**Visualization:** Angela Elam.

**Writing – original draft:** Angela Elam, Dennis Akrobetu, Tochukwu Ndukwe, David Sidhom, Shikha Marwah.

**Writing – review & editing:** Angela Elam, Dennis Akrobetu, Tochukwu Ndukwe, David Sidhom, Patrice M. Hicks, Shikha Marwah, Brian C. Stagg, Paul P. Lee, Josh D. Stein, Paula Anne Newman-Casey.

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
