## [Decision Letter · Decision Letter 0]

Dec 02 2024

Dear Dr. Elam,

Thank you for submitting your manuscript to PLOS ONE. After careful consideration, we feel that it has merit but does not fully meet PLOS ONE’s publication criteria as it currently stands. Therefore, we invite you to submit a revised version of the manuscript that addresses the points raised during the review process.

**The manuscript could benefit from the additional explanations requested by the reviewers. **

We look forward to receiving your revised manuscript.

Kind regards,

Swarnali Goswami

Academic Editor

PLOS ONE

2. Please amend your manuscript to include your abstract after the title page.

3. Please include captions for your Supporting Information files at the end of your manuscript, and update any in-text citations to match accordingly. Please see our Supporting Information guidelines for more: http://journals.plos.org/plosone/s/supporting-information.

4.  Please include a caption for table 1 and 2.

Reviewers' comments:

Reviewer's Responses to Questions

**Comments to the Author**

1. Is the manuscript technically sound, and do the data support the conclusions?

Reviewer #1: Yes

Reviewer #2: Yes

2. Has the statistical analysis been performed appropriately and rigorously?

Reviewer #1: Yes

Reviewer #2: No

3. Have the authors made all data underlying the findings in their manuscript fully available?

Reviewer #1: Yes

Reviewer #2: Yes

4. Is the manuscript presented in an intelligible fashion and written in standard English?

Reviewer #1: Yes

Reviewer #2: Yes

Reviewer #1: The authors address a key research question that highlights the disparities in Medicaid and Medicare reimbursements for ophthalmic procedures. The findings of the study have major health policy implications in terms of differences that exist for the same procedure being incentivized differently by Medicaid and Medicare as well as the differential acceptance rate for patients covered under one over the other health insurance.

See a few of my comments and questions below:

Methods:

1. Please provide additional information on what was the data source? Cite if possible. Is it directly from CMS?

2. Does this data also cover Medicare managed care (Part C) and Medicaid managed care patients?

3. How were dual-eligible patients (Medicaid and Medicare eligible) captured?

Discussion:

1. Based on the findings, what could be the reason for the large variation in CV especially for intravitreal injection administration across different states? Typically intravitreal injections are given for conditions such as diabetic retinopathy and age-related macular degeneration. Is there a possibility that in some of the states the rate of reimbursement is low due to a skewed population with lower proportion of patients having these conditions in comparison to the national average?

2. Some states have a higher proportion of Medicaid-eligible population than other states, similarly this variation can exist at a more granular level within a county. Due to a higher Medicaid population in certain regions, the ophthalmologists could be highly disincentivized to provide care to Medicaid especially if Medicaid reimbursement is low. Is there a negative correlation between (a) Medicaid population geographic distribution and rate of Medicaid reimbursement and (b) Medicaid population geographic distribution and rate of acceptance of Medicaid?

3. Many of the procedures/surgeries selected in your study are typically performed on an elderly population. Having said that, Medicaid population is on average younger than Medicare population. In this case, if an elderly individual qualifying for Medicaid and Medicare may preferentially get treated for the same condition with the same procedure with the clinician leveraging the patient’s Medicare status to secure a higher reimbursement rate. Is this a possibility? If yes, are there state-specific regulations on managing dual-eligible individuals or is it purely on the discretion of the clinician?

4. Since individual states determine Medicaid reimbursements while CMS determines Medicare reimbursement. Does individual state Medicaid perceive the same procedure differently than CMS from a complexity and effort standpoint or is the reimbursement also associated with the state Medicaid limited budget allocation?

Reviewer #2: Summary:

The authors quantify the differences between Medicare and Medicaid reimbursement for specific ophthalmic procedures and by different states. This study also correlates the ophthalmologist acceptance of Medicaid patients with these reimbursement differences by region. Overall, Medicaid reimbursement was found to have more variability and be less than Medicare reimbursement with a few exceptions. I think this study adds to the literature on disparities between Medicare and Medicaid in ophthalmology and lays the foundation for future work of studying outcomes related to these disparities. Thank you for the opportunity to review this article. I enjoyed reading it. I have a few comments and suggestions.

Major Points

- I think an independent t-test would be more appropriate over a paired t-test since I would consider Medicare and Medicaid samples to be independent from each other. Whereas a paired t-test would be used to compare reimbursements in Medicaid between 2023 and 2024, for example.

- Was any formal correlation calculated for the association between Medicaid reimbursement and physician acceptance of Medicaid patients?

Minor Points

- In the last sentence in the first introduction paragraph, low income is mentioned twice.

- “in New Jersey with Medicaid paying $998.78 less then Medicare” – then should be than

- I think the figures are helpful in visualizing the data. I would suggest that figures 3 and 4 plot the percentage of acceptance rates rather than the counts so that the bars can be easily compared across regions.

- Figure 4 is a little blurry - could a higher resolution figure be provided?

**Do you want your identity to be public for this peer review?** For information about this choice, including consent withdrawal, please see our Privacy Policy

Reviewer #1: No

Reviewer #2: **Yes: ** Jennifer Toth Harris

---

## [Author Response · Author response to Decision Letter 1]

11 Apr 2025

Please see responses to reviewer comments below. These were also included in our cover letter for your review. Thank you for the opportunity to improve our manuscript!

--Done

2. Please amend your manuscript to include your abstract after the title page.

--Done

3. Please include captions for your Supporting Information files at the end of your manuscript, and update any in-text citations to match accordingly. Please see our Supporting Information guidelines for more: http://journals.plos.org/plosone/s/supporting-information. Caption included for Supplemental Table 1 and cited in the text.

--Done

4. Please include a caption for table 1 and 2.

--Captions for Tables 1 and 2 are including in on the documents.

--We did not find any retracted references.

Reviewers' comments:

Reviewer's Responses to Questions

Comments to the Author

1. Is the manuscript technically sound, and do the data support the conclusions?

Reviewer #1: Yes

Reviewer #2: Yes

2. Has the statistical analysis been performed appropriately and rigorously?

Reviewer #1: Yes

Reviewer #2: No

3. Have the authors made all data underlying the findings in their manuscript fully available?

Reviewer #1: Yes

Reviewer #2: Yes

4. Is the manuscript presented in an intelligible fashion and written in standard English?

Reviewer #1: Yes

Reviewer #2: Yes

5. Review Comments to the Author

Reviewer #1: The authors address a key research question that highlights the disparities in Medicaid and Medicare reimbursements for ophthalmic procedures. The findings of the study have major health policy implications in terms of differences that exist for the same procedure being incentivized differently by Medicaid and Medicare as well as the differential acceptance rate for patients covered under one over the other health insurance.

See a few of my comments and questions below:

Methods:

1. Please provide additional information on what was the data source? Cite if possible. Is it directly from CMS?

--Supplemental Table 1 includes the data sources for both Medicare and Medicaid. The Medicare data comes directly from CMS. The Medicaid comes from each state.

2. Does this data also cover Medicare managed care (Part C) and Medicaid managed care patients?

--The reimbursement rates included in this study represent “straight” Medicare and Medicaid rates. The managed care reimbursement rates, on average, are comparable to standard plan rates.

3. How were dual-eligible patients (Medicaid and Medicare eligible) captured?

--This study does not include patients or patient coverage, but instead explores the differences between the two insurance types’ physician reimbursement rates.

Discussion:

1. Based on the findings, what could be the reason for the large variation in CV especially for intravitreal injection administration across different states? Typically intravitreal injections are given for conditions such as diabetic retinopathy and age-related macular degeneration. Is there a possibility that in some of the states the rate of reimbursement is low due to a skewed population with lower proportion of patients having these conditions in comparison to the national average?

--Thank you for this question. It is difficult to know why and how states come to a decision regarding reimbursement rates for various procedures. If we look at diabetes for example, the national prevalence diabetes varies from approximately 7% to 14% across the U.S. However, these prevalence rates do not appear to correlate with reimbursement rates.

2. Some states have a higher proportion of Medicaid-eligible population than other states, similarly this variation can exist at a more granular level within a county. Due to a higher Medicaid population in certain regions, the ophthalmologists could be highly disincentivized to provide care to Medicaid especially if Medicaid reimbursement is low. Is there a negative correlation between (a) Medicaid population geographic distribution and rate of Medicaid reimbursement and (b) Medicaid population geographic distribution and rate of acceptance of Medicaid?

--Thank you for this insightful question. We hope to explore these variations at a more granular level in a future publication. The following is included in the Discussion section: “While our data do not allow us to demonstrate a statistically significant correlation, we do find that acceptance rates of Medicaid vary by region and appear to correlate with reimbursement rates. Regions with lower Medicaid acceptance rates also have greater dollar difference in Medicaid and Medicare reimbursement. In the Midwest region, the Medicaid acceptance rate is relatively higher, and we notice the reimbursement difference between Medicaid and Medicare is lowest in that region, meaning the Medicaid reimbursement is higher (closer to that of Medicare).”

3. Many of the procedures/surgeries selected in your study are typically performed on an elderly population. Having said that, Medicaid population is on average younger than Medicare population. In this case, if an elderly individual qualifying for Medicaid and Medicare may preferentially get treated for the same condition with the same procedure with the clinician leveraging the patient’s Medicare status to secure a higher reimbursement rate. Is this a possibility? If yes, are there state-specific regulations on managing dual-eligible individuals or is it purely on the discretion of the clinician?

--We imagine this is possible. In most cases, for patients that are dually insured by Medicare and Medicaid, Medicare pays first, then Medicaid provides supplemental coverage as needed.

4. Since individual states determine Medicaid reimbursements while CMS determines Medicare reimbursement. Does individual state Medicaid perceive the same procedure differently than CMS from a complexity and effort standpoint or is the reimbursement also associated with the state Medicaid limited budget allocation?

--Thank you for this question. It is difficult to know why and how states come to a decision regarding reimbursement rates for various procedures and that information was not readily available to use. This is certainly a question that would be helpful to explore, particularly from an advocacy/policy standpoint. It is outside of the scope of this paper, but something to consider in future publications looking at the data more granularly.

Reviewer #2: Summary:

The authors quantify the differences between Medicare and Medicaid reimbursement for specific ophthalmic procedures and by different states. This study also correlates the ophthalmologist acceptance of Medicaid patients with these reimbursement differences by region. Overall, Medicaid reimbursement was found to have more variability and be less than Medicare reimbursement with a few exceptions. I think this study adds to the literature on disparities between Medicare and Medicaid in ophthalmology and lays the foundation for future work of studying outcomes related to these disparities. Thank you for the opportunity to review this article. I enjoyed reading it. I have a few comments and suggestions.

Major Points

- I think an independent t-test would be more appropriate over a paired t-test since I would consider Medicare and Medicaid samples to be independent from each other. Whereas a paired t-test would be used to compare reimbursements in Medicaid between 2023 and 2024, for example.

--Thank for this suggestion. Our rationale for using a paired t-test in these analyses is as follows: In this study we have focused on comparing the difference between the Medicare and Medicaid reimbursement payments across states. Since the unit of study is state, and we have tested the null hypothesis that the differences between the two reimbursements are equal across the states in the country, it is appropriate to use the paired t-test. These data are paired by state, and our results show that there is evidence of a statistically significant difference in these payments across the states for specific ophthalmic procedures. We have reviewed your insightful question with our biostatisticians and believe that the appropriate analyses were conducted given the research questions posed.

- Was any formal correlation calculated for the association between Medicaid reimbursement and physician acceptance of Medicaid patients?

--Yes, we did conduct analyses to identify any correlation between Medicaid reimbursement rates and acceptance. These analyses were not statistically significant. We include the following in the discussion section: “While our data do not allow us to demonstrate a statistically significant correlation, we do find that acceptance rates of Medicaid vary by region and appear to correlate with reimbursement rates. Regions with lower Medicaid acceptance rates also have greater dollar difference in Medicaid and Medicare reimbursement. In the Midwest region, the Medicaid acceptance rate is relatively higher, and we notice the reimbursement difference between Medicaid and Medicare is lowest in that region, meaning the Medicaid reimbursement is higher (closer to that of Medicare).”

Minor Points

- In the last sentence in the first introduction paragraph, low income is mentioned twice.

-- Thank you. We have updated the sentence to read “…provides medical coverage primarily to low-income individuals, children, and people with disabilities.”

- “in New Jersey with Medicaid paying $998.78 less then Medicare” – then should be than

--Correction made. Thank you.

- I think the figures are helpful in visualizing the data. I would suggest that figures 3 and 4 plot the percentage of acceptance rates rather than the counts so that the bars can be easily compared across regions.

--Great suggestion, thank you. Initially we included counts for transparency, but appreciate your suggestion and have updated the figures to plot percentages and include counts in the notes.

- Figure 4 is a little blurry - could a higher resolution figure be provided?

--Thank you for pointing this out. As mentioned above, we have updated Figures 3 and 4.

6. PLOS authors have the option to publish the peer review history of their article (what does this mean?). If published, this will include your full peer review and any attached files.

Do you want your identity to be public for this peer review? For information about this choice, including consent withdrawal, please see our Privacy Policy.

Reviewer #1: No

Reviewer #2: Yes: Jennifer Toth Harris

---

## [Editor Report · Decision Letter 1]

Disparities in Medicaid and Medicare Physician Reimbursements for Ophthalmic Procedures

PONE-D-24-11096R1

Dear Dr. Elam,

We’re pleased to inform you that your manuscript has been judged scientifically suitable for publication and will be formally accepted for publication once it meets all outstanding technical requirements.

Kind regards,

Swarnali Goswami

Academic Editor

PLOS ONE
---

## [Editor Report · Acceptance letter]

PONE-D-24-11096R1

PLOS ONE

Dear Dr. Elam,

I'm pleased to inform you that your manuscript has been deemed suitable for publication in PLOS ONE. Congratulations! Your manuscript is now being handed over to our production team.

Kind regards,

on behalf of

Dr. Swarnali Goswami

Academic Editor

PLOS ONE